# Fluence-dependent degradation of fibrillar type I collagen by 222 nm far-UVC radiation

Antonia Kowalewski [1,2], Nancy R. Forde [1,2]*

**1** Department of Physics, Simon Fraser University, Burnaby, BC, Canada, **2** Department of Molecular Biology and Biochemistry, Simon Fraser University, Burnaby, BC, Canada

* nforde@sfu.ca

**Data Availability Statement:** All relevant data are within the manuscript and its Supporting information files.

**Funding:** NRF: Natural Sciences and Engineering Research Council of Canada (NSERC) Discovery Grant, RGPIN-2020-04680 https://www.nserc-

## Abstract

For more than 100 years, germicidal lamps emitting 254 nm ultraviolet (UV) radiation have been used for drinking-water disinfection and surface sterilization. However, due to the carcinogenic nature of 254 nm UV, these lamps have been unable to be used for clinical procedures such as wound or surgical site sterilization. Recently, technical advances have facilitated a new generation of germicidal lamp whose emissions centre at 222 nm. These novel 222 nm lamps have commensurate antimicrobial properties to 254 nm lamps while producing few short- or long-term health effects in humans upon external skin exposure. However, to realize the full clinical potential of 222 nm UV, its safety upon internal tissue exposure must also be considered. Type I collagen is the most abundant structural protein in the body, where it self-assembles into fibrils which play a crucial role in connective tissue structure and function. In this work, we investigate the effect of 222 nm UV radiation on type I collagen fibrils *in vitro*. We show that collagen's response to irradiation with 222 nm UV is fluence-dependent, ranging from no detectable fibril damage at low fluences to complete fibril degradation and polypeptide chain scission at high fluences. However, we also show that fibril degradation is significantly attenuated by increasing collagen sample thickness. Given the low fluence threshold for bacterial inactivation and the macroscopic thickness of collagenous tissues *in vivo*, our results suggest a range of 222 nm UV fluences which may inactivate pathogenic bacteria without causing significant damage to fibrillar collagen. This presents an initial step toward the validation of 222 nm UV radiation for internal tissue disinfection.

## Introduction

Surgical-site infections are a leading cause of health care-associated infections worldwide, affecting up to one-third of surgical patients in low- and middle-income countries and up to 10% of surgical patients in Europe [1]. The majority of these infections are caused by antimicrobial-resistant pathogens [1]. Ultraviolet C (UVC) radiation, defined as electromagnetic radiation of wavelengths 200 to 280 nm, effectively inactivates both drug-sensitive and drug-resistant bacteria [2–11]. While this property has been exploited for water and surface disinfection for over 100 years, the use of UVC radiation for medical purposes has remained limited

crsng.gc.ca/index_eng.asp The funders had no role in study design, data collection and analysis, decision to publish, or preparation of the manuscript.

due to the mutagenic effects of 254 nm UV, the most readily available UVC wavelength [12, 13]. More recently, however, technical advances have facilitated the production of shorter-wavelength 222 nm "far-UVC" lamps which have commensurate anti-bacterial properties to 254 nm UV lamps but do not cause mutagenic DNA damage upon external human skin exposure [2, 14, 15]. Given its high level of absorbance by proteins, it is proposed that far-UVC radiation is absorbed by the protein-rich anucleate cells of the stratum corneum before it can reach the nucleated cells of the lower epidermis and induce DNA damage [2]. Some studies suggest that far-UVC radiation is also unable to penetrate the cytoplasm of individual unprotected eukaryotic cells [16, 17]. In bacterial cells, however, due to a small cell volume and lack of a protective stratum corneum, far-UVC radiation remains cytotoxic [2].

So far, investigations into the human safety of far-UVC radiation have focused on external skin exposure [2, 15, 18–22]. Despite this, far-UVC radiation is actively being proposed for open-wound sterilization and surgical-site disinfection [6, 14, 19, 23, 24]. To determine the safety of far-UVC radiation for such exposures, it is necessary to investigate its effect on the body's internal tissues. In this study, we investigate the effect of 222 nm far-UVC radiation on type I collagen fibrils, the main protein component of the extracellular matrix (ECM) and connective tissues including tendons, ligaments, skin, bone, and cartilage [25, 26].

Type I collagen fibrils are composed of laterally associated type I collagen molecules, each of which is a heterotrimer of two $\alpha_1$ and one $\alpha_2$ polypeptide chains arranged in a 300 nm-long right-handed triple helix [27, 28] (Fig 1). Fully formed fibrils range from 30 to 300 nm in diameter and are stabilized by non-covalent interactions between collagen molecules [29–31]. In the ECM, collagen fibrils form a network which can be modelled experimentally by a fibrillar gel [32] (Fig 1).

Previous studies have investigated the effect of 254 nm UVC radiation on collagen. However, to the authors' knowledge, this study presents the first investigation into the effect of 222 nm far-UVC radiation on type I collagen fibrils.

## Materials and methods

### Sample preparation

Collagen fibrils were formed by combining rat tail tendon-derived acid-soluble type I collagen (5 mg/mL in 20 mM acetic acid) (CultrexRat Collagen I, R&D Systems) with 400 mM HEPES + 0.02% sodium azide and 1 M NaCl at room temperature to reach final concentrations of 1 mg/mL collagen, 100 mM HEPES + 0.005% sodium azide, and 270 mM NaCl. Samples were added to a clear-bottom 96-well plate (Nunc MicroWell 96-Well Optical-Bottom Plate with Polymer Base, Thermo Fisher Scientific) (60 μL per well) or flowed into a 0.175 or 1.2 mm-

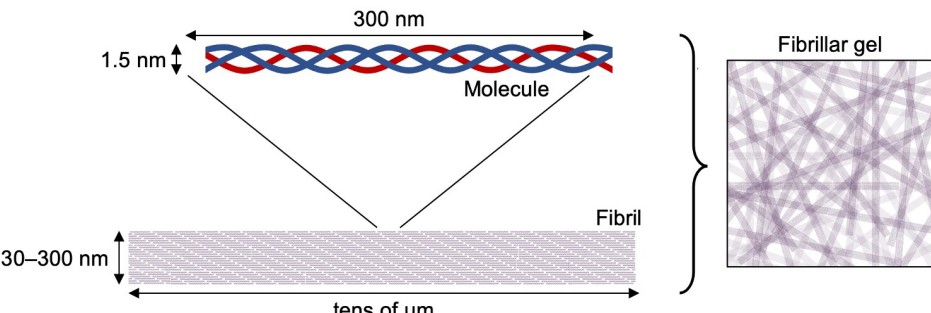

**Fig 1. Structure of a type I collagen fibrillar gel.** Blue lines represent $\alpha_1$ chains and red lines represent $\alpha_2$ chains.

thick microscope sample chamber and sealed with nail polish. Sample chambers consist of a 1.0 mm–thick ge124-grade quartz microscope slide (75875–470, VWR) and a 0.13 to 0.16 mm–thick borosilicate glass coverslip (12–542B, Fisherbrand) separated by two parallel stacks of double-sided tape (Tesa) spaced approximately 5 mm apart (S1 Fig). All fibrillar collagen samples were left to polymerize at room temperature for at least four hours prior to irradiation and analysis. Fibril formation under these conditions was confirmed by monitoring the optical density of the solution at 347 nm with a BioTek Synergy HTX microplate reader (Agilent) with 100 μL per well (S2 Fig). Molecular collagen samples were prepared in the same way as fibrillar collagen samples (1 mg/mL in 100 mM HEPES + 0.005% sodium azide and 270 mM NaCl) but were irradiated immediately following neutralization, before fibril formation could occur.

Bovine serum albumin (BSA) samples were prepared by diluting lyophilized BSA powder (Sigma Aldrich) to 1 mg/mL in ddH$_2$O and adding to a clear-bottom 96-well plate (60 μL per well).

## UV irradiation

Sample irradiation was performed using a LILY Handheld Personal Far UV Disinfection Light from UV Can Sanitize Corp. (North Vancouver, BC, Canada) after a minimum 45-minute warm-up time. The lamp's emission spectrum is tightly peaked around 222 nm (S3 Fig). Before each irradiation, the irradiance (power per unit area, E) at the surface of the sample (1 cm from the lamp) was determined using a Molectron PM5100 Laser Power Meter with a 1 cm$^2$ paper mask. From the irradiance, the exposure time (t) necessary to achieve the desired fluence (H) was calculated using the equation H = E×t. For samples in sample chambers, irradiation was performed through a quartz microscope slide with 67% transmittance (as measured using a Molectron PM5100 Laser Power Meter), and exposure times were adjusted accordingly. Exposure times ranged from 4 s to 23 min.

## SDS–PAGE

Samples destined for SDS-PAGE were irradiated in a clear-bottom 96-well plate (see above) with 60 μL per well, corresponding to a sample depth of about 1.9 mm. Each irradiated sample was transferred to a 0.5 mL tube with an equal volume of 2X reducing Laemmli buffer (4% sodium dodecyl sulfate, 10% β-mercaptoethanol, 125 mM Tris–HCl, 20% glycerol, 0.002% bromophenol blue) and heated at 95˚C for 10 minutes. Samples were loaded into a discontinuous SDS–PAGE system (5% stacking gel, 6 or 8% resolving gel, 3 or 6 μg protein per well) and run in a Bio-Rad Mini-PROTEAN Tetra Vertical Electrophoresis Cell at 50 mA for approximately 45 minutes alongside a prestained protein ladder (PageRuler Prestained Protein Ladder, 10 to 180 kDa, Thermo Fisher Scientific). Gels were stained for 1 h in a Coomassie solution (0.1% Coomassie brilliant blue R-250, 40% ethanol, 10% acetic acid) and de-stained overnight in 10% ethanol and 7.5% acetic acid. Following de-staining, gels were imaged with an Olympus model C-5060 Wide Zoom camera in a UV transilluminator at 302 nm (High Performance UV Transilluminator, UVP).

## Microscopy

Samples destined for microscopy were irradiated in a 0.175 or 1.2 mm-thick sample chamber (see above). Bright-field microscopy was performed using an Olympus IX83 inverted light microscope with an ORCA-Spark camera (Hamamatsu). Due to the optical resolution limit, only collagen fibrils, not monomeric collagen molecules, are visible in a conventional light microscope. No image post-processing was performed.

## Results and discussion

Collagen fibrils in 0.175 mm-thick sample chambers were irradiated with 1000, 5000, or 10,000 mJ/cm$^2$ of far-UVC radiation. Exposure to 1000 mJ/cm$^2$ of far-UVC radiation resulted in significant collagen fibril degradation and disruption of the fibrillar network (Fig 2A). Exposure to 5000 and 10,000 mJ/cm$^2$ of far-UVC radiation caused apparently complete collagen fibril degradation and loss of the fibrillar network.

To determine whether fibril degradation was a result of depolymerization or chain scission, irradiated fibrils were analyzed by SDS–PAGE. Type I collagen molecules produce five characteristic bands on an SDS–PAGE gel: $\alpha_1$ and $\alpha_2$ bands with a 2:1 ratio, $\beta_{11}$ (two cross-linked $\alpha_1$ chains) and $\beta_{12}$ (cross-linked $\alpha_1$ and $\alpha_2$ chains) bands with a 1:2 ratio, and a $\gamma$ band (three cross-linked $\alpha$ chains). Following exposures of 5000, 10,000, and 20,000 mJ/cm$^2$ of far-UVC radiation, SDS-PAGE analysis revealed a decrease in intensity of collagen $\alpha$, $\beta$, and $\gamma$ bands with increasing far-UVC fluence (Fig 2B). This indicates that far-UVC radiation-induced fibril degradation involves polypeptide chain scission. Furthermore, the absence of new low molecular weight bands in the lanes of the irradiated samples suggests that far-UVC radiation-induced chain scission is random and/or results in fragments smaller than 40 kDa. These results are consistent with existing literature on the effect of 221 and 254 nm UV radiation on type I collagen, which reports collagen chain scission at high fluences [33–38]. A subset of the aforementioned studies also describe UV-induced collagen cross-linking, and suggest that cross-linking dominates at low fluences and chain scission at high fluences [34, 35, 37]. Cross-linking would be seen in an SDS–PAGE gel by the appearance of higher molecular weight bands. We observe no evidence of cross-linking following far-UVC exposure of collagen fibrils at fluences down to approximately 360 mJ/cm$^2$ (S6 Fig).

To elucidate whether far-UVC radiation-induced chain scission is dependent on protein structure or sequence, both molecular type I collagen and bovine serum albumin (BSA) were

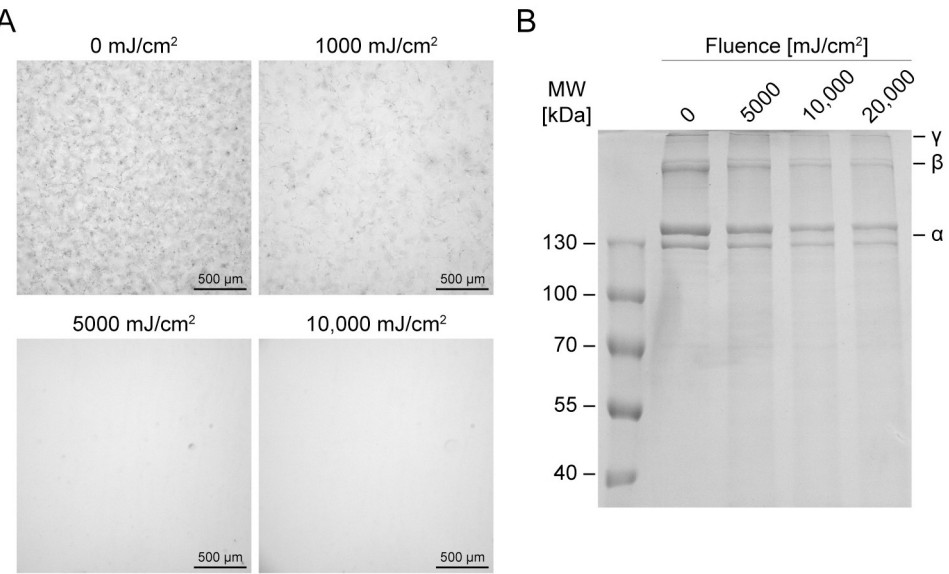

**Fig 2. Collagen fibrils before and after irradiation with far-UVC.** A: Collagen fibrils exposed to 1000, 5000, and 10,000 mJ/cm$^2$ of far-UVC radiation and imaged under a light microscope at 10X magnification. B: Collagen fibrils exposed to 5000, 10,000, and 20,000 mJ/cm$^2$ of far-UVC radiation and run on an SDS–PAGE gel (8% resolving gel) with 6 μg protein per well.

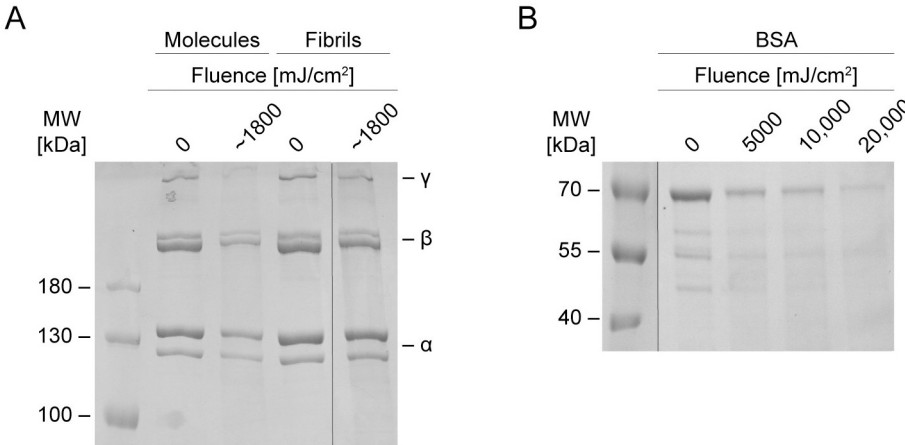

**Fig 3. SDS-PAGE gels of molecular collagen, fibrillar collagen, and BSA before and after irradiation with far-UVC.** A: Collagen molecules and fibrils exposed to equal fluences of far-UVC radiation and run on a 6% resolving gel with 3 μg protein per well. The black vertical line indicates where two portions of the original gel were spliced together to remove irrelevant lanes. B: BSA molecules irradiated with 5000, 10,000, and 20,000 mJ/cm² of far-UVC radiation and run on an 8% resolving gel with 6 μg protein per well. The black vertical line indicates where two portions of the original gel were spliced together to remove irrelevant lanes.

irradiated and subjected to SDS–PAGE. As shown in Fig 3A, there was a greater loss of intensity of collagen α, β, and γ bands for collagen irradiated in molecular form than collagen irradiated with the same fluence in fibrillar form, indicating that the former underwent more chain scission than the latter. However, the optical density at 222 nm of a fibrillar collagen solution is greater than that of a molecular collagen solution due to increased light scattering [31]. Thus, the observed effect may be a function of increased far-UVC attenuation in the fibrillar solution rather than a structure-dependent effect of collagen molecules on the outside of fibrils shielding molecules on the inside. Fig 3B shows that irradiation of BSA with 5000, 10,000, and 20,000 mJ/cm² of far-UVC resulted in a pattern of fluence-dependent band disappearance similar to that observed for fibrillar collagen (Figs 2B and 3B). We conclude that exposure to far-UVC radiation can lead to chain scission in both fibrous and globular proteins, and that the fluence-dependent effects are not specific to collagen.

To further assess how optical density influences fibrillar collagen network disruption by far-UVC radiation, we prepared fibrillar collagen samples with two different thicknesses for irradiation. The optical density (OD) of a dilute sample at wavelength λ is $OD_\lambda = (\mu_{a,\lambda} + \mu_{s,\lambda}) d$, where $\mu_{a,\lambda}$ is the absorption coefficient, $\mu_{s,\lambda}$ is the scattering coefficient, and $d$ is the sample thickness. Since the absorption coefficient of all samples used in this study is constant (all samples have the same chemical composition), we can modulate OD by changing either the scattering coefficient $\mu_{s,\lambda}$ or the sample thickness $d$. The decrease in OD of a molecular collagen solution compared to a fibrillar solution is due to a smaller scattering coefficient for molecular collagen, as described above. Upon increasing OD by increasing sample thickness from 0.175 mm to 1.2 mm, we observed a significant reduction in fibril degradation, with intact fibrils present in the 1.2 mm sample even after exposure to approximately 2500 mJ/cm² of far-UVC radiation (Fig 4). Because the extent of scattering depends on sample thickness, scattering by fibrils in the thicker sample leads to attenuation of the 222 nm light, and less radiation penetrates to the deeper regions of the sample chamber. In a clinical scenario, any fibril damage that were to occur as a result of far-UVC exposure would likely be restricted to the top layers of tissue.

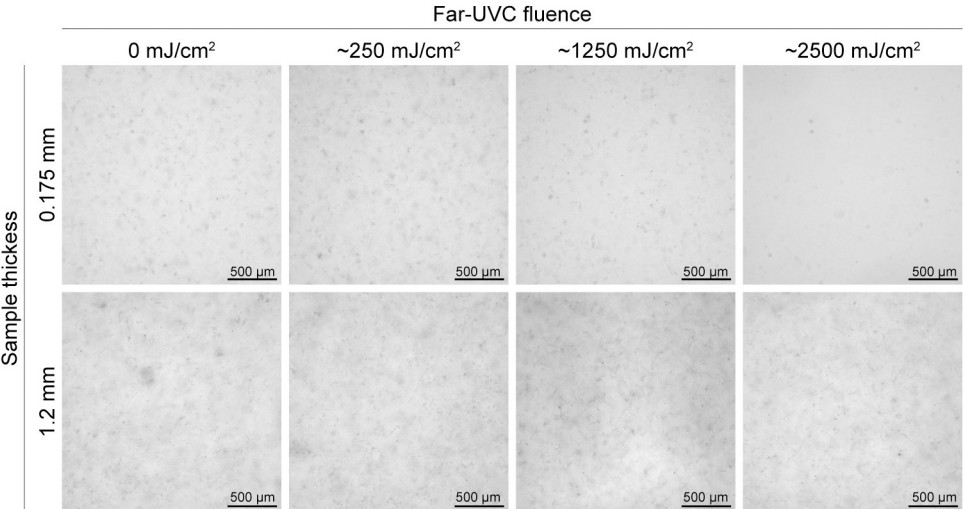

**Fig 4. Fibrillar collagen samples of different thicknesses before and after irradiation with far-UVC.** Collagen fibrils were irradiated with increasing fluences of far-UVC radiation and imaged under a light microscope at 10X magnification. Top row corresponds to a thin sample, bottom row corresponds to a thick sample.

While these results provide an initial validation of the safety of far-UVC radiation for extracellular matrix and connective tissue exposure, further analysis is needed in more tissue-realistic systems. The fibrillar collagen samples used in this study lack cells and matrix components beyond collagen, as well as intermolecular collagen cross-links present *in vivo* [39]. Furthermore, the fluences employed here are greater than those which would be used clinically. Previous work has established that far-UVC radiation is an effective antibacterial agent at fluences below 100 mJ/cm$^2$ [40, 41]. It will be important to investigate whether low fluences of far-UVC radiation induce collagen cross-link formation [34, 35, 37], as increased ECM cross-linking is associated with tissue fibrosis and impaired wound healing [42, 43]. Far-UVC radiation is also ozone-generating, which poses a potential health hazard for both patients and healthcare workers [44]. In addition to being toxic when inhaled, ozone can generate free radicals and cause oxidation of biological molecules [45]. In our experiments, exposure to far-UVC radiation caused gas bubbles to form within sealed sample chambers, with higher fluences leading to more and larger bubbles (S3 Fig). However, the amount of gas produced was minimal even at a far-UVC fluence of 1000 mJ/cm$^2$, 10-fold greater than that needed to inactivate most bacteria. Finally, this study did not investigate whether long exposures to low-intensity light yield different results than short exposures to high-intensity light (to achieve the same fluence). The optimization of these parameters will be crucial to inform the design of far-UVC disinfection units and clinical protocols.

## Conclusions

In this work, we showed that the response of type I collagen fibrils to far-UVC radiation is dependent on both far-UVC fluence and fibrillar network thickness (Table 1). At low fluences and high thicknesses—the most representative of a clinical setting—fibrillar collagen is resistant to significant degradation by far-UVC radiation. Only at fluences many times greater than the bactericidal dose did far-UVC radiation induce random polypeptide chain cleavage in both collagen and BSA. We conclude that germicidal far-UVC lamps should continue to be investigated as promising candidates for the prevention of surgical-site infections.

**Table 1. Summary of fluence-dependent observations.**

| Fluence [mJ/cm²] | Sample type | Sample thickness [mm] | Observations | | Reference |
|---|---|---|---|---|---|
| | | | SDS–PAGE | Microscopy | |
| 250 | col. fibrils | 0.175 | | no degradation | Fig 4 |
| | | 1.2 | | no degradation | Fig 4 |
| 360 | col. fibrils | 2 | slight degradation | | S6 Fig |
| 900 | col. fibrils | 2 | slight degradation | | S6 Fig |
| 1000 | col. fibrils | 0.175 | | significant degradation | Fig 2A |
| 1250 | col. fibrils | 0.175 | | slight degradation | Fig 4 |
| | | 1.2 | | no degradation | Fig 4 |
| 1800 | col. fibrils | 2 | slight degradation | | Fig 3A |
| | col. molecules | 2 | significant degradation | | Fig 3A |
| | BSA | 2 | slight degradation | | S6 Fig |
| 2500 | col. fibrils | 0.175 | | complete degradation | Fig 4 |
| | | 1.2 | | no degradation | Fig 4 |
| 5000 | col. fibrils | 0.175 | | complete degradation | Fig 2A |
| | | 2 | slight degradation | | Fig 2B |
| | BSA | 2 | significant degradation | | Fig 3B |
| 10,000 | col. fibrils | 0.175 | | complete degradation | Fig 2A |
| | | 2 | slight degradation | | Fig 2B |
| | BSA | 2 | significant degradation | | Fig 3B |
| 20,000 | col. fibrils | 2 | significant degradation | | Fig 2B |
| | BSA | 2 | significant degradation | | Fig 3B |

Col., type I collagen; BSA, bovine serum albumin.

## Supporting information

**S1 Fig. Experimental configurations in which collagen samples were irradiated with relevant volumes and dimensions labelled.** A: Single well of a 96-well plate. B: Microscopy sample chamber. C: Digital photograph of a representative microscopy sample chamber.
(TIF)

**S2 Fig. Optical density at 347 nm as a function of time following initiation of collagen fibril formation.** Error bars represent the standard error of the mean from three replicate measurements.
(TIF)

**S3 Fig. Emission spectra of a LILY Handheld Personal Far UV Disinfection Light.** Data courtesy of Viso Systems (Copenhagen, Denmark). Figure adapted from Viso Systems.
(TIF)

**S4 Fig. Microscopy sample chambers containing collagen fibrils before and after irradiation with far-UVC.** Gas bubbles are increasingly prevalent at higher fluences.
(TIF)

**S5 Fig. Original SDS-PAGE gel image for Figs 2B and 3B.**
(TIF)

**S6 Fig. Original SDS-PAGE gel image for Fig 3A.**
(TIF)

## Acknowledgments

The authors thank Mehrdad Behnami and Rea-Jayne Smith of UV Can Sanitize Corp. for generously supplying the far-UVC lamp, Jens Lassen and Gary Leach for lending power meters, and Jody Tao and other members of the Forde Lab for their research insights and suggestions.

## Author Contributions

**Conceptualization:** Antonia Kowalewski.

**Formal analysis:** Antonia Kowalewski.

**Funding acquisition:** Nancy R. Forde.

**Investigation:** Antonia Kowalewski.

**Methodology:** Antonia Kowalewski.

**Supervision:** Nancy R. Forde.

**Writing – original draft:** Antonia Kowalewski.

**Writing – review & editing:** Antonia Kowalewski, Nancy R. Forde.

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
