## [Decision Letter · Decision Letter 0]

13 Oct 2023

PONE-D-23-28770Fluence-dependent degradation of fibrillar type I collagen by 222 nm far-UVC radiationPLOS ONE

Dear Dr. Forde,

Thank you for submitting your manuscript to PLOS ONE. After careful consideration, we feel that it has merit but does not fully meet PLOS ONE’s publication criteria as it currently stands. Therefore, we invite you to submit a revised version of the manuscript that addresses the points raised during the review process.

We look forward to receiving your revised manuscript.

Kind regards,

Amitava Mukherjee, ME, Ph.D.

Academic Editor

PLOS ONE

“The authors thank Mehrdad Behnami and Rea-Jayne Smith of UV Can Sanitize Corp. for generously supplying the far-UVC lamp, Jens Lassen and Gary Leach for lending power meters, and Jody Tao and other members of the Forde Lab for their research insights and suggestions. This work was funded by the Natural Sciences and Engineering Research Council of Canada (NSERC) through a Discovery Grant to NRF.”

“NRF: Natural Sciences and Engineering Research Council of Canada (NSERC) Discovery Grant, RGPIN-2020-04680

https://www.nserc-crsng.gc.ca/index_eng.asp

Reviewers' comments:

Reviewer's Responses to Questions

**Comments to the Author**

1. Is the manuscript technically sound, and do the data support the conclusions?

Reviewer #1: Yes

Reviewer #2: Yes

2. Has the statistical analysis been performed appropriately and rigorously? 

Reviewer #1: Yes

Reviewer #2: N/A

3. Have the authors made all data underlying the findings in their manuscript fully available?

Reviewer #1: Yes

Reviewer #2: Yes

4. Is the manuscript presented in an intelligible fashion and written in standard English?

PLOS ONE does not copyedit accepted manuscripts, so the language in submitted articles must be clear, correct, and unambiguous. Any typographical or grammatical errors should be corrected at revision, so please note any specific errors here

Reviewer #1: Yes

Reviewer #2: Yes

5. Review Comments to the Author

Reviewer #1: This is a well written article that provides much needed information on the safety of 222 nm light use. This is an area receiving a lot of attention right now and the information provided by the authors provides new insights into the safetyfor medical applications.

Reviewer #2: 1. How does this study on collagen differ from the previous studies on collagen and UVC exposure?

2. Have any controls set up to account for other variables that could impact the optical density affecting fibrillar collagen network disruption?

3. For clarity, the authors can tabulate or graphically depict the effects of exposure time, fluence, and collagen thickness.

4. It is recommended that the authors include digital photographs of their sample chambers.

5. The authors say that "Thicker samples act as a barrier for radiation attenuation and overall fibril damage", if that is the case, what should be the threshold thickness of collagen

6. Figure 2B of the study indicates no cross-linking in the SDS-PAGE gel after far-UVC exposures of 5000 mJ/cm2 or higher. Are there any additional techniques to better understand potential cross-linking?

7. "Figure 2B shows no evidence of cross-linking following far-UVC exposures of 5000 mJ/cm2" - Is fluence parameter the sole determining factor for cross-linking?

8. The results on collagen fibrils lack intermolecular cross-links in vivo. How might this affect the overall conclusions and their applicability in a real-world context for germicidal UVC exposure?

9. If far-UVC radiation has the capability to generate ozone. Then, is it possible for the presence of ozone to result in any interactive or synergistic effects when combined with far-UVC radiation on collagen or other tissue components?

10. The conclusion

(a) mentions "low fluences" and "high thicknesses," but it does not specify quantities or ranges. Quantitative data or a reference to the specifications will strengthen the conclusion.

(b) suggests germicidal far-UVC lamps for surgical-site infection prevention. Recommending specific additional studies, long-term studies, or scalability of such lamps in surgical settings would provide depth.

6. PLOS authors have the option to publish the peer review history of their article (what does this mean?). If published, this will include your full peer review and any attached files.

Reviewer #1: No

Reviewer #2: No

---

## [Author Response · Author response to Decision Letter 0]

28 Nov 2023

We have uploaded a cover letter that includes the response to editorial comments, and a response to reviewers document that contains responses to each of the reviewer comments. Please see those two uploaded documents for the responses.

---

## [Decision Letter · Decision Letter 1]

10 Dec 2023

Fluence-dependent degradation of fibrillar type I collagen by 222 nm far-UVC radiation

PONE-D-23-28770R1

Dear Dr. Forde,

We’re pleased to inform you that your manuscript has been judged scientifically suitable for publication and will be formally accepted for publication once it meets all outstanding technical requirements.

Kind regards,

Amitava Mukherjee, ME, Ph.D.

Academic Editor

PLOS ONE

Additional Editor Comments (optional):

Reviewers' comments:

Reviewer's Responses to Questions

**Comments to the Author**

1. If the authors have adequately addressed your comments raised in a previous round of review and you feel that this manuscript is now acceptable for publication, you may indicate that here to bypass the “Comments to the Author” section, enter your conflict of interest statement in the “Confidential to Editor” section, and submit your "Accept" recommendation.

Reviewer #2: All comments have been addressed

2. Is the manuscript technically sound, and do the data support the conclusions?

Reviewer #2: Yes

3. Has the statistical analysis been performed appropriately and rigorously? 

Reviewer #2: N/A

4. Have the authors made all data underlying the findings in their manuscript fully available?

Reviewer #2: Yes

5. Is the manuscript presented in an intelligible fashion and written in standard English?

Reviewer #2: Yes

6. Review Comments to the Author

Reviewer #2: The authors have addressed all the comments and incorporated the necessary details in the revised version. Hence, the manuscript can be accepted for publication

7. PLOS authors have the option to publish the peer review history of their article (what does this mean?). If published, this will include your full peer review and any attached files.

Reviewer #2: No

---

## [Editor Report · Acceptance letter]

18 Dec 2023

PONE-D-23-28770R1 

PLOS ONE

Dear Dr. Forde, 

I'm pleased to inform you that your manuscript has been deemed suitable for publication in PLOS ONE. Congratulations! Your manuscript is now being handed over to our production team.

Kind regards, 

on behalf of

Professor Dr. Amitava Mukherjee 

Academic Editor

PLOS ONE